# Acute Effects of Short-Term Massage Procedures on Neuromechanical Contractile Properties of Rectus Femoris Muscle

**DOI:** 10.3390/medicina60010125

**Published:** 2024-01-10

**Authors:** Miloš Dakić, Vladimir Ilić, Lazar Toskić, Sasa Duric, Jožef Šimenko, Milan Marković, Milivoj Dopsaj, Ivan Cuk

**Affiliations:** 1Faculty of Sport and Physical Education, University of Belgrade, 11000 Belgrade, Serbia; miloshdakic@gmail.com (M.D.); vladimir.ilic@fsfv.bg.ac.rs (V.I.); milivoj.dopsaj@fsfv.bg.ac.rs (M.D.); 2Faculty of Sport and Physical Education, University of Priština in Kosovska Mitrovica, 38218 Leposavić, Serbia; lazar.toskic@pr.ac.rs (L.T.); milan.markovic@pr.ac.rs (M.M.); 3Faculty of Sport, University “Union–Nikola Tesla”, 11070 Belgrade, Serbia; 4Liberal Arts Department, American University of the Middle East, Egaila 54200, Kuwait; sasa.duric@aum.edu.kw; 5Faculty of Sport, University of Ljubljana, 1000 Ljubljana, Slovenia; jozef.simenko@fsp.uni-lj.si

**Keywords:** tensiomyography, isometric fatigue, massage modalities, muscle recovery, training

## Abstract

*Background and Objectives*: In many sports, maintaining muscle work at an optimal level despite fatigue is crucial. Therefore, it is essential to discover the most efficient way of recovery. This study aimed to evaluate and compare the acute effects of four different recovery methods on muscle neuromechanical properties. *Materials and Methods*: The research was conducted using a randomized, quasi-experimental, repeated-measures design. Fourteen healthy and active male students of the Faculty of Sport and Physical Education (age 25.1 ± 3.9 years) were included in this study. The tensiomyography was used to evaluate muscle responses after four different types of short-term recovery methods (passive rest, percussive mechanical, vibro-mechanical, and manual massage) on the rectus femoris muscle on four occasions: baseline, post fatigue, post recovery and prolonged recovery. *Results*: The ANOVA revealed that muscle fatigue decreased maximal vertical muscle displacement (Dm) and muscle contraction time (Tc) in post fatigue compared to the baseline. The most important finding shows that only the vibro-mechanical massage resulted in an increase in Tc in the prolonged recovery compared to the post fatigue (*p* = 0.028), whereas only manual massage showed no differences in Dm from the baseline in post-recovery (*p* = 0.148). Moreover, both manual and vibro-mechanical massages increased Dm and Tc in prolonged recovery, indicating no differences from the baseline (all *p* > 0.05), thus showing signs of muscle recovery. Percussion mechanical massage and passive rest did not show indices of muscle recovery. *Conclusions*: Manual massage could induce immediate positive changes in Dm by reducing muscle stiffness. In addition, vibro-mechanical and manual massage improved muscle tissue by rapidly returning Dm and Tc values to baseline at prolonged recovery measurement (5 min after the fatigue protocol). These findings can benefit sports practitioners, and physical therapists in developing the best recovery method after muscle fatigue.

## 1. Introduction

Long-term, dedicated, hard work is a prerequisite for serious athletic success. However, training and competing at a high level brings a certain amount of fatigue and stress that, if not reduced, can negatively affect the athlete’s body or outcome [1]. Passive recovery is often not enough for the athlete to recover fully [2]. For this reason, athletes must have a quality recovery process that restores and improves muscle function in addition to expertly guided training [3]. In many sports, athletes have little time to prepare for a game during competition when they are tired. After short breaks or a time-out, maintaining muscle work at an optimal level despite fatigue is critical. This makes it essential for athletes to recover as quickly and efficiently as possible. For this reason, it is up to researchers to find the most efficient way of recovery. There are numerous training-related, psychological, and medical recovery methods [4]. Massage is a method that is frequently used for this very purpose: to recover an athlete and prepare them for the following efforts. It is a mechanical manipulation of soft tissues through rhythmically applied movements and pressure [5,6]. There are many types of massage: manual massage, acupressure massage, rolling massage, vibro-massage, percussion massage, hydro-massage, and electro-massage, all aimed at improving the recovery process [7,8].

It is crucial to find sufficiently sensitive methods to detect the changes caused by the effect of massages. There is a lack of suitable, more sensitive systems for measuring the contractile properties of muscles after fatigue. This led researchers to search for alternative methods to provide more accurate insight into the contractile mechanisms. In addition to surface electromyography, neuromuscular function has been studied using ultrasound [9,10], magnetic resonance imaging (MRI) [11], transcranial magnetic stimulation (TMS) [12], mechanomiography (MMG) [13,14], myotonic technology (MyotonPRO) [15], and tensiomyography (TMG). In recent studies, tensiomyography is a valid and reliable method for examining the neuromechanical contractile properties of muscles [16]. Tensiomyography represents an essential link in the sports and training system. It is based on the response of the muscles to an appropriate electrical stimulus, enabling the detection of neuromechanical changes during external muscle electrostimulation using a sensor that records the radial displacement of the muscle belly [17,18]. This sensor provides highly accurate information on muscle contractile velocity, muscle stiffness, and, indirectly, functional and lateral symmetry, which plays a vital role in injury prevention, rehabilitation, and training processes [19].

In recent years, the influence and effects of massage on the neuromechanical contractile properties of muscles, often used as a means of recovery, have become an increasingly interesting topic of study [20,21,22,23,24]. Although the studies mentioned above examined the effects of different types of massage (manual, mechanical, and foam roller massage), only one study aimed to compare them and determine possible differences [24]. Their findings showed that there was no difference between the mentioned treatments. In actual sports circumstances, athletes have different pauses during competition when coaches or team staff can intervene directly. Depending on the sports modality, these pauses include timeouts, technical timeouts, substitutions, and intervals between parts, sets, or quarters. However, only substitutions and timeouts can be freely managed by the coach, and these two tools are considered very important for managing the team during the competition [25]. Therefore, it can be regarded as one of the most essential tools in team sport management. It allows coaches to provide direct instructions to their players [26] and physicians to attend to them. In the light of applicable sports science, there is a significant lack of studies focusing on massage usage in these short periods.

Based on the abovementioned problem, this research aims to evaluate the effects of four different types of short-term recovery methods (manual, vibro-mechanical, and percussive mechanical massage and passive rest) on the neuromechanical contractile properties of the rectus femoris muscle, measured using the TMG method after an isometric fatigue protocol. We hypothesized that any massage treatment immediately after the fatigue protocol would significantly positively affect the neuromechanical contractile properties of the treated muscle group compared to passive rest. The results of this study could provide crucial information on the influence of different short-term massage therapies on muscle recovery, which could lead to further development of sports training and the achievement of peak results.

## 2. Materials and Methods

### 2.1. Ethical Approval

This study was approved by the Institutional Review Board of the Faculty of Sport and Physical Education, the University of Belgrade (IRB: 02-2650/23-1). This study was conducted following recognized ethical standards according to the Declaration of Helsinki adopted in 1964 and revised in 2013.

### 2.2. Participants

Fourteen healthy and active male students of the Faculty of Sport and Physical Education, aged between 20 and 30, were included in this study (25.1 ± 3.9 years, 78.6 ± 8.9 kg, 181.3 ± 6.0 cm, 23.9 ± 2.1 kg∙m^−2^). The primary criterion for including subjects in the study was no history of neuromuscular diseases or musculoskeletal injuries. For the duration of the experiment, subjects were not involved in any massage process. They maintained their daily routine regarding physical activity and were asked not to take any dietary supplements or medications. All participants signed a written consent to participate in the study.

### 2.3. Experimental Protocol

The research was conducted using a randomized, repeated-measures design in which participants were required to visit the research laboratory on five different days/sessions, separated by a rest period of 5–7 days [27]. On the first day, in the morning, subjects were familiarized with the experimental protocol and the custom-made isometric dynamometer. In addition to body height (Seca 220, Seca, Hamburg, Germany) and weight (Seca 769, Seca, Hamburg, Germany), maximal voluntary isometric contraction (MVIC) was also measured. The research protocol for the remaining four visits was based on TMG measurements, fatigue, and recovery procedures. TMG measurements were performed at rest, immediately after a fatigue procedure, immediately after a recovery procedure, and five minutes after fatigue (PRT) to determine any prolonged effects of the treatment. Four groups of recovery procedures were used in the study: passive rest, percussion mechanical massage, vibro-mechanical massage, and manual massage, and they were applied in randomized order (Figure 1). All measurements were performed under the same conditions and by the same experienced researcher.

### 2.4. Experimental Procedures

#### 2.4.1. TMG Measurements

The measurement of neuromechanical contractile properties of muscles was performed on the rectus femoris muscle using the tensiomyography method (TMG-BMC, Ljubljana, Slovenia), following all procedures recommended by the manufacturer (20). This muscle was chosen as it assists in knee extension, hip flexion, and stabilization of the pelvis on the femur when bearing [28], and these are the most common movements in sports. The subjects were in a relaxed supine position, with the angle in the knee joint at 120° [29]. Immediately before the electrodes were attached, they were asked to perform a voluntary contraction to determine the position for placement of the TMG measurement sensor by visual and palpatory methods. The electrode was marked and placed at the site of the greatest vertical displacement of the muscle belly [22]. Two self-adhesive electrodes (Pals Platinum, model 895220 with multistick gel, Axelgaard Manufacturing Co. Ltd., five cm^2^) were placed there. They emitted an electrical impulse proximally and distally at a distance of 55 to 60 mm from the marked site [29]. The sensor to detect changes and obtain data was placed between the electrodes (GK40, Panoptik, Ljubljana, Slovenia) (Figure 2).

Neuromechanical contractile properties were assessed using an electrical pulse of 40 mA for 1 ms. The impulses were increased proportionally by 20 mA every 10 s until the moment when any additional muscle response to the increase in electrostimulation disappeared [24,30]. The two best scores were used for further statistical analysis. The TMG method provides us with different variables, such as muscle contraction time (Tc), delayed muscle contraction time (Td), muscle relaxation time (Tr), duration of contraction (Ts), and maximal vertical muscle displacement (Dm) [31]. The variables that were monitored and analyzed in our study were contraction time (Tc), the time required to reach from 10% to 90% of the maximum vertical movement of the muscle, and maximum vertical muscle movement during electrical involuntary stimulation (Dm), as the most reliable variables of tensiomyography in assessing muscle fatigue [32]. Tc reflects the rate of contraction from the onset to the end of muscle contraction and is related to the rate of force generation, whereas Dm measures the movement of the muscle belly expressed in millimeters and is considered an indirect measure of muscle tone or stiffness [33]. Tc is a physiological component of neuromuscular function. On the other hand, Dm is considered an anatomical component determined by the number and type of muscle fibers recruited by the electrical stimulus [34].

#### 2.4.2. MVIC, Warm-Up, and Fatigue Procedures

MVIC, warm-up, and fatigue procedures were carried out in an adjustable custom-made chair specifically designed for this type of testing. Participants assumed a comfortable seated position and were restrained with straps across the chest, hips, and thighs. The rigid strap and force sensor were placed approximately 2 cm above the lateral malleolus of the right foot so that the knee joint was flexed to 120° (full knee extension was 180°) (Figure 3).

The signals from the force sensor (“CZL302”, GSCS Electronic Measuring Technology Co., Ltd. Guangdong, China) were acquired using commercially available software (“Isometrics, Sports Medical Solutions,” Belgrade, Serbia) version 4.0., with a sampling frequency of 1000 Hz and filtered with a second-order Butterworth low-pass filter (5 Hz). For MVIC measurement, subjects were asked to perform three maximal contractions with 2 min of rest in between. Contractions lasted 3 to 5 s, and subjects were verbally encouraged during their trials. The highest MVIC value was selected as the reference value for the upcoming procedures. Participants had visual feedback of their contraction forces on a screen in front of them the entire time. For warm-up, subjects performed three 40 s isometric contractions at 40% of their MVIC with a 30 s rest in between. The fatigue procedure consisted of five contractions at 40% of MVIC until task failure, with one minute rest between contractions [35]. Task failure was defined as the participant’s inability to maintain force within 10% of the target force. A horizontal boundary line was placed beneath the required contraction intensity to facilitate visual feedback.

#### 2.4.3. Recovery Treatments

In addition to manual massage as a typical means of recovery, mechanical massages in electromassage devices are increasingly used in sports practice [36]. The largest number of sports workers used Hypervolt^®^ (54%) and Theragun^®^ (38%) in their work, indicating the quality of these devices [36]. Therefore, we decided to use these types of recovery methods.

All treatments were performed by a therapist with extensive experience and lasted one minute. The applied massages followed the direction of the fibers of the rectus femoris muscle using moderate force and rapid movement, gliding along the muscle belly from origin to insertion. During the treatment, the participants gave subjective feedback on the intensity of the massage pressure [37]. It should also be noted that the same technique or move can often be given different names in different styles (e.g., longitudinal friction, deep twitch, or muscle shaping are the same), so massage therapists with different training may not realize when they are applying the same technique [38].

Four techniques are performed in manual massage (MM) for 15 s each (effleurage, friction, tapping, and vibration, respectively). In the effleurage treatment (Figure 4, panel A), the movements are performed with the palms. The friction (Figure 4, panel B) involved flat motions with the forearms, moving the tissue over the underlying structures. Tapping (Figure 4, panel C) refers to continuous rhythmic movements in which the hands come into contact with the tissue, similar to playing percussion instruments. Vibrations (Figure 4, panel D) included vibrating or oscillatory rhythmic movements to release tension. In order to accomplish an optimal effect with this type of massage, the region of the body where the vibrations are performed should be relaxed. The main difference between tapping and vibration is that vibrations do not break contact with the tissue [39,40,41].

For the vibro-mechanical (VM) massage (Figure 4, panel E), we used the already mentioned Hypervolt^®^. Of the three possible working speeds, we used the middle one with 2600 vibrations per minute and a working amplitude of 13 mm.

For the percussive mechanical (PM) massage (Figure 4, panel F), we used the Theragun^®^. An operating speed of 2400 strokes per minute and a ball extension with a working amplitude of 16 mm were used [36].

Finally, participants also have passive rest (PR) as a form of recovery.

### 2.5. Statistical Analysis

Descriptive statistics were calculated as mean and standard deviation (SD). First, the Kolmogorov–Smirnov test revealed that none of the dependent variables deviated significantly from the normal distribution. A repeated-measures ANOVA was performed for the main effect (measurement time point) to compare the dependent variables obtained at the four different time points. Additionally, two-way ANOVA (between within) was performed to assess the differences between recovery procedures. In case of significant differences in individual procedures, a Bonferroni post hoc test was performed. Along with the ANOVAs, Eta squared (ŋ^2^) was also calculated, with effect sizes of 0.01, 0.06, and above 0.14 considered small, medium, and large, respectively [42]. Alpha was set at *p* < 0.05. All statistics were performed in the SPSS 26 (IBM, Armonk, NY, USA).

## 3. Results

The results presented in Table 1 depict the descriptive statistics of the measured TMG parameters at four time points in four different recovery procedures.

Figure 5 shows a graphical representation of the Tc and Dm results at four time points and recovery procedures. In almost all procedures, Tc and Dm values decreased after the fatigue protocol and gradually increased again.

Table 2 presents the results of the repeated measures and two-way ANOVA (between within) to determine the influence of the short-term recovery procedures and differences between the recovery procedures (Recovery procedure x Measurement time point). It can be concluded that there were significant differences between TMG measurements within each individual recovery procedure (Wilks’ Lambda = 0.387, F = 6.260, *p* = 0.019, η2 = 0.612, on average). At the same time, there were no significant interaction effects (Wilks’ Lambda = 0.873, F = 0.778, *p* = 0.637, η2 = 0.044); that is, there were no significant differences between the measurement procedures.

Table 3 contains the pairwise comparison results, i.e., the comparison between TMG parameters in all four recovery procedures. Significant differences between baseline and post-fatigue measurements exist in all four recovery procedures for both Tc (*p* = 0.038, on average) and Dm (*p* = 0.018, on average). In addition, differences were found between the baseline measurement and almost all post-recovery measurements (except Dm in manual massage) (*p* = 0.016, on average) and between the baseline measurement and the measurement taken five minutes after the fatigue protocol for Tc and Dm of the passive rest (*p* = 0.028, *p* = 0.005, respectively) and Theragun^®^ (*p* = 0.001 and *p* = 0.009, respectively) procedures. The most important result is that there are no significant differences in the measured TMG parameters of any recovery procedure between the post-fatigue and post-recovery measurements (*p* = 0.942, on average). The only significant difference between the post-fatigue measurement and the measurement taken five minutes after the fatigue protocol is in the Tc of the Hypervolt^®^ procedure (*p* = 0.028). Finally, there are no significant differences in both Tc and Dm for any recovery protocol between the post-recovery measurement and the measurement taken five minutes after the fatigue protocol (*p* = 0.603).

## 4. Discussion

The present study aimed to evaluate and compare the acute effects of four different short-term recovery methods on the neuromechanical contractile muscle properties, i.e., contraction time and the maximum displacement of the rectus femoris muscle using the TMG method after isometrically induced muscle fatigue. The most important finding of this study was that manual massage could cause immediate positive changes in Dm by reducing muscle stiffness. In addition, vibro-mechanical and manual massage improved muscle tissue by rapidly returning Dm and Tc values to baseline at the PRT measurement. Moreover, the results of our study indicate that muscle fatigue affected both Dm and Tc parameters.

To observe the effects of fatigue, it is necessary to select and apply an exercise with sufficient intensity to induce neuromechanical changes in the muscle. According to the literature, muscle fatigue should lead to an increase in Tc and a decrease in Dm. An increase in Tc is associated with decreased contraction velocity, which should lead to impaired muscle power. On the other hand, a decrease in Dm leads to higher muscle stiffness during contraction. This means that during the fatigue state, cellular structures and conductive properties of membranes are impaired, and the efficiency of neuromuscular contractile capacity is reduced [43]. The recovery procedures should bring Tc and Dm values as close as possible to initial values as possible. In this regard, it is essential to point out that the Dm parameter, according to several authors, can show the most valuable data when examining the contractile properties of muscles through TMG [44,45,46]. On the other hand, Tc values should be taken with caution. Despite the theoretical assumptions that Tc values should increase with fatigue, our results showed an opposite trend. However, as mentioned earlier, that is in line with most of the studies conducted recently. In most previous studies, Dm showed a significant decreasing trend, implying muscle fatigue increases tension and stiffness after exercise [24,43,47,48,49]. Only a few studies showed no changes [50,51]. However, the Tc response to fatiguing exercise yielded more controversial results. It remained constant [43,51], increased [52], decreased [24,47,49], or showed different outcomes depending on the muscle group [48,50].

This study demonstrated that the protocol consisting of five isometric contractions at 40% of MVC until failure with one minute of rest successfully induced these changes by decreasing both Tc and Dm values (Table 1). Muscle stiffness was increased to overcome the fatigue caused by isometric testing until failure. An increase in tendinous muscle rigidity can explain this finding. In our case, as mentioned, Tc values were also decreased after the fatigue protocol. Although this contradicts the theoretical assumptions, it can be explained by facilitating muscle function by improving the contractile component of force production. Decreased Dm may lead to a faster twitch response and a higher rate of force production, thus shortening muscle contraction time [53]. One explanation for this could be the reduced rate of actin–myosin bond separation [54,55]. This phenomenon may be considered as a way to store more energy generated during exercise by increasing muscle stiffness rather than decreasing muscle activity as muscle fatigue increases [56]. The Tc reduction can also be explained by the fact that the acute exercise used in our study did not cause significant muscle fatigue due to the post-exercise post-activation potentiation (PAP). Post-activation potentiation is a physiological phenomenon related to acute neuromuscular and performance improvements, through which acute muscle force output is enhanced due to contractile history [57,58]. It can compensate for fatigue during endurance exercise, increase the rate of force development, and thus improve speed and power performance [59]. TMG is a valid and sensitive method to detect the PAP effects only in type II muscle fibers, with a sensor mounted directly over the muscle belly [60]. Monitoring PAP and its effect on TMG was not the aim of this study so future research could pay more attention to this phenomenon.

The recovery methods used in this study were passive rest, percussive mechanical massage, vibro-mechanical massage, and manual massage, and their use was time limited to one minute (time-out duration). These methods should help reduce post-exercise tension, increase blood circulation, enhance recovery, stimulate the exercising muscles, and prepare them for maximum performance [61]. As their use increases, it is necessary to find valid and sensitive methods for evaluating the mechanisms of muscle activity after massage treatment [62]. This is the first study to fulfill these methodological inconsistencies among the protocols concerning the duration of applied massages. Immediately after the fatigue protocol and TMG measurement, we performed one of the four recovery procedures. It should be mentioned that the main purpose of the recovery procedures was to return the Tc and Dm values to the initial baseline values. The short-term messages used in this study yielded some interesting findings. Regarding Tc, none of the applied recovery procedures had immediate effects, meaning the differences mostly remained significant after the treatment (*p* < 0.05). However, manual and vibro-mechanical massage affected muscle properties five minutes after the fatigue protocol compared to baseline TMG measurement at rest (*p* = 0.287 and *p* = 0.399, respectively). It was also found that after the fatigue procedure, only the vibro-mechanical massage resulted in faster recovery in PRT measurement compared to the other recovery procedures (*p* = 0.028).

On the other hand, manual massage had an immediate positive effect on muscle stiffness after the fatigue protocol compared to baseline TMG measurement (*p* = 0.148). Manual and vibro-mechanical massage also affected muscle properties five minutes after the fatigue protocol compared to baseline TMG measurement at rest (*p* = 0.205 and *p* = 0.105, respectively). Interestingly, manual massage resulted in the fastest return of muscle stiffness to baseline values. The experience and feeling of the therapist could play a crucial role in this. Moreover, the last technique performed during manual massage was the vibration technique, so it can be assumed that it could contribute to reducing stiffness. Percussion mechanical massage and passive rest did not affect muscle stiffness (*p* < 0.05). An increase in local stiffness may affect adjacent tissues via collagenous connective tissue [63]. It is possible that when penetrated through muscle, applied vibrations had more positive results than percussions. One possible mechanism for these findings could be that vibration therapy can stimulate more muscle receptors, resulting in increased blood flow, cutaneous vascular conduction, and improved oxygen delivery to the muscle [61]. It has also been reported that applied vibrations led to mechanical oscillatory motions associated with increased intramuscular temperature [64]. Vibration may play an essential role in muscle recovery.

### Limitations

This study needs to acknowledge some limitations. All respondents were male students, so no possible differences were observed between men and women, nor between the participants’ physical fitness levels (recreationists, professional athletes). Also, we measured only one type of muscle fatigue (isometric) in one muscle group (rectus femoris); we did not try to create game situations (different sports) and then evaluated the muscle response. In addition, subjective feelings of muscle pain were not analyzed in order to use the perceptual scales of recovery. It would be advisable to try to create effective field game-based situations, as well as massage models with precise massage techniques (studying other massage techniques) and their modalities (i.e., different therapeutic tools, different attachment heads on handheld devices, different intensity of applied pressure, different speed of movement, etc.) that could help optimize training and recovery programs. In addition, further studies are needed to determine the periods in which contractile properties return to baseline values to investigate both the short- and long-term effects of massage and the physiological and psychological responses that occur after fatigue and applied treatments. This way, science could provide exercise physiologists and coaches with essential clues for better training periodization.

## 5. Conclusions

The ability to recover after intense training or competitive bouts is essential to maintain or even increase performance in subsequent efforts. Developing adequate recovery systems for the most effective and efficient preparation of athletes for competitions is one of the most important goals in today’s sport. Our goal was to create a situation as similar as possible to practical sports conditions, where the therapist has very little time to help the athlete overcome the current state of fatigue.

The main finding of this study was that manual massage induced a significant immediate positive change in muscle displacement (Dm) by reducing muscle stiffness. Additionally, the vibro-mechanical and manual massage affected Tc and Dm five minutes after fatigue protocol, indicating an acute response of neuromechanical properties. However, only the vibro-mechanical massage applied led to a faster return of Tc values to the baseline in the PRT measurement. As an additional value of our study, it is important to note that our protocol can be used as a muscle fatigue indicator when attempting to induce fatigue through isometric exercise because it was able to cause changes in Tc and Dm in the rectus femoris muscle by decreasing both parameters. Thus, TMG has proven to be a reliable tool in assessing this type of muscle fatigue. To conclude, vibro-mechanical and manual massage may be helpful recovery methods to improve muscle tissue after isometrically induced fatigue. These findings can benefit sports practitioners, and physical therapists in applying the best recovery method after muscle fatigue. As a result, improved muscles’ neuromuscular capabilities might lead to better sports results while reducing the risk of injuries.

## Figures and Tables

**Figure 1 medicina-60-00125-f001:**
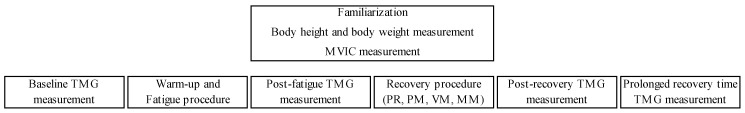
Schematic depiction of the experimental protocol.

**Figure 2 medicina-60-00125-f002:**
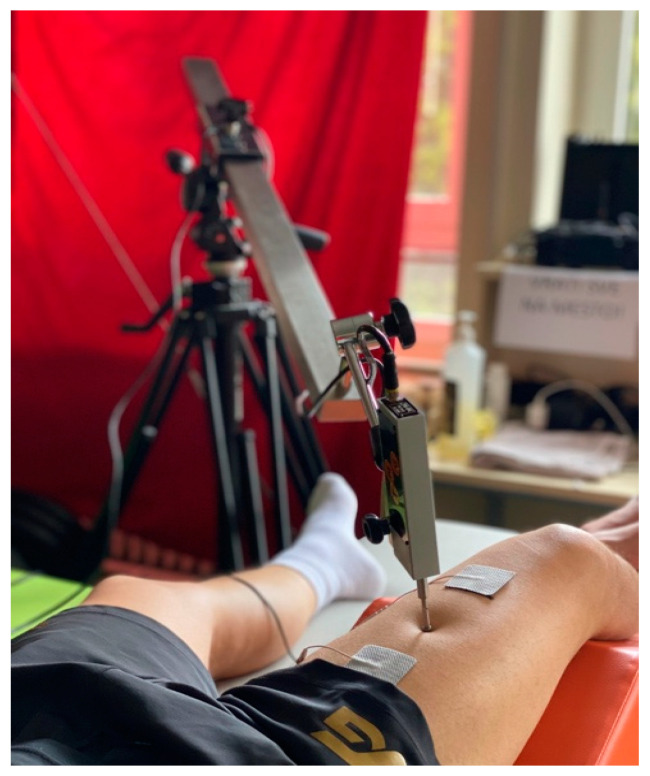
TMG measurement procedure.

**Figure 3 medicina-60-00125-f003:**
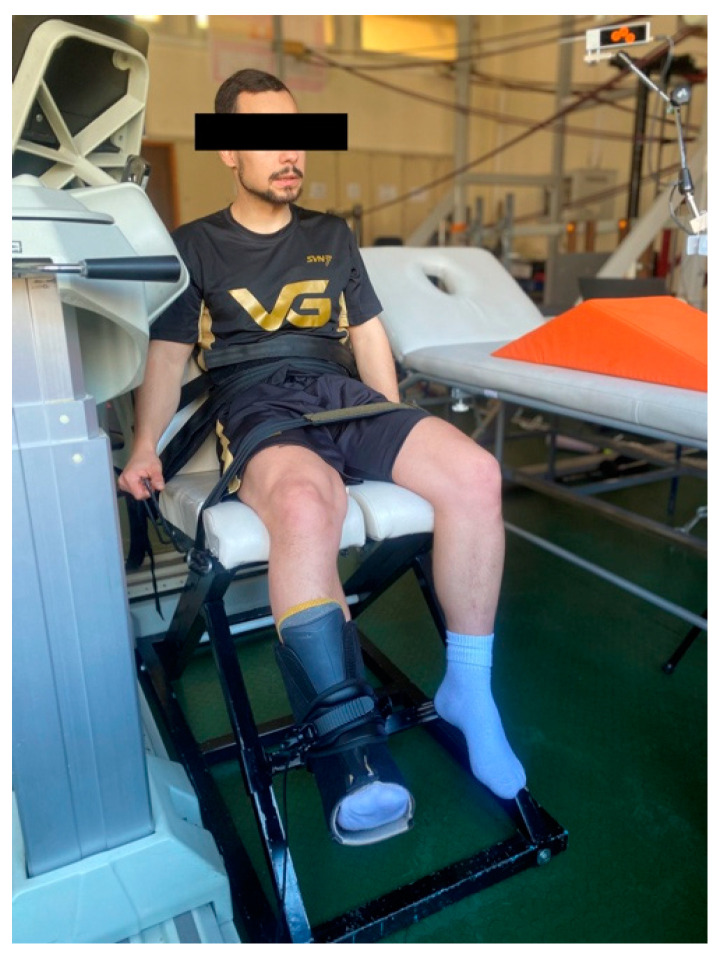
Adjustable custom-made chair for MVIC, warm-up, and fatigue procedure.

**Figure 4 medicina-60-00125-f004:**
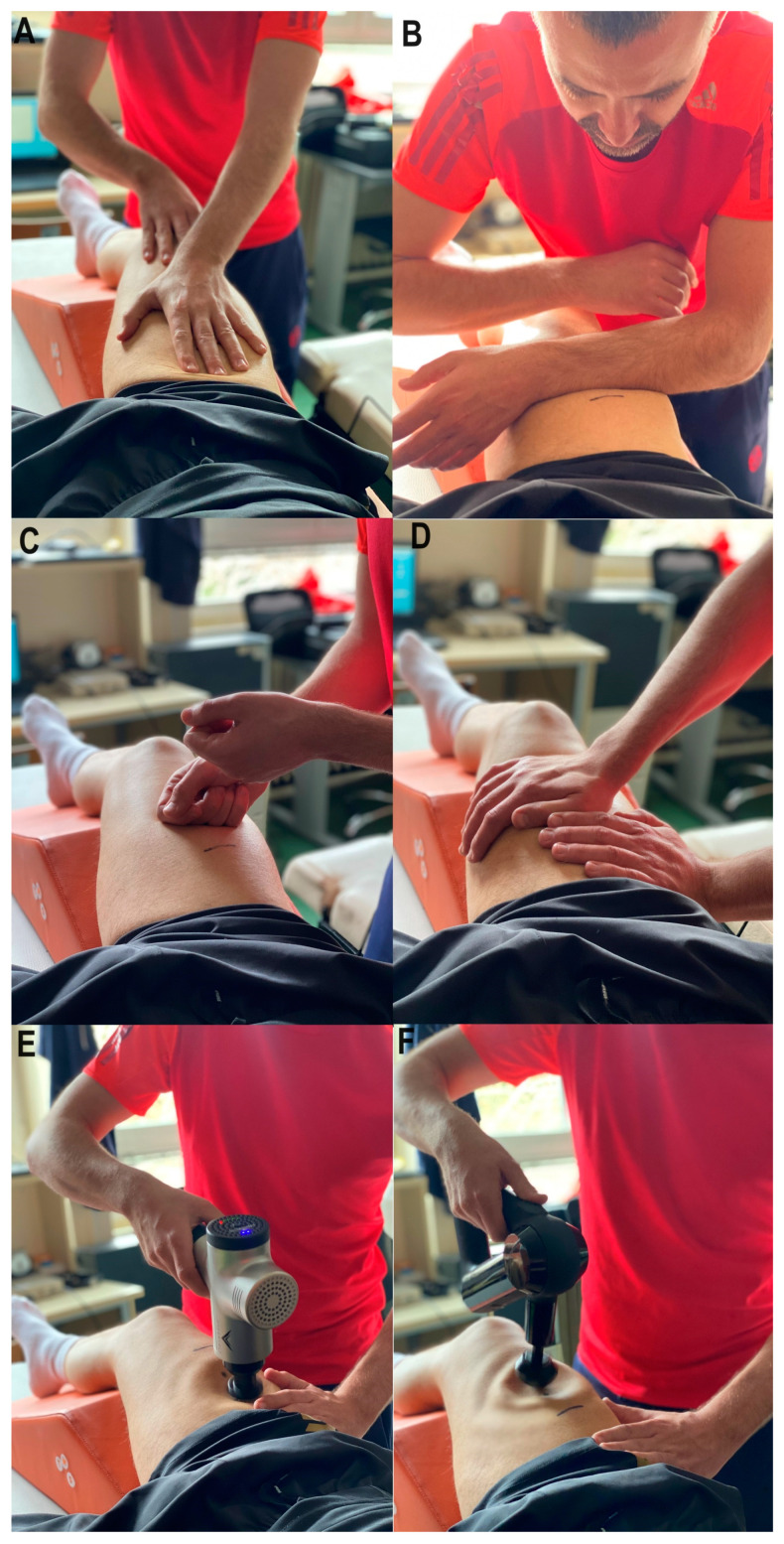
Applied recovery treatments. Panel (**A**–**D**) depicting manual massage techniques (effleurage, friction, tapping, and vibration, respectively); panel (**E**) depicting vibro-mechanical massage; and panel (**F**) percussive mechanical massage.

**Figure 5 medicina-60-00125-f005:**
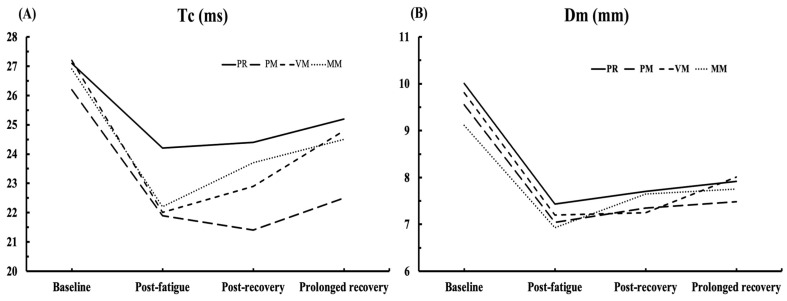
Changes in Tc (panel (**A**)) and Dm (panel (**B**)) through four time-point measurements. PR—passive rest, PM—percussive-mechanical massage (Theragun^®^), VM—vibro-mechanical massage (Hypervolt^®^), MM—manual massage.

**Table 1 medicina-60-00125-t001:** Descriptive values (Mean ± SD) of TMG parameters Tc (ms) and Dm (mm) in four measurements in four different recovery procedures.

	Baseline Measurement		Post-Fatigue Measurement		Post-Recovery Measurement		Prolonged Recovery Time Measurement	
	Tc	Dm	Tc	Dm	Tc	Dm	Tc	Dm
**PR**	27.1 ± 4.1	10.0 ± 2.7	24.2 ± 5.1	7.4 ± 2.5	24.4 ± 4.1	7.7 ± 2.4	25.2 ± 3.8	7.9 ± 2.5
**PM**	26.2 ± 4.8	9.5 ± 3.5	21.9 ± 6.0	7.4 ± 3.1	21.4 ± 4.0	7.3 ± 3.1	22.5 ± 4.3	7.5 ± 3.1
**VM**	27.2 ± 5.0	9.8 ± 3.3	22.0 ± 5.5	7.2 ± 3.1	22.9 ± 5.0	7.2 ± 2.5	24.8 ± 5.8	8.0 ± 2.7
**MM**	26.9 ± 4.6	9.1 ± 2.6	22.2 ± 4.0	6.9 ± 2.9	23.7 ± 3.9	7.6 ± 2.8	24.5 ± 4.9	7.7 ± 2.5

Abbreviation: Tc—muscle contraction time; Dm—maximal radial muscle displacement; PR—passive rest; PM—percussive mechanical (Theragun^®^); VM—vibro-mechanical massage (Hypervolt^®^); MM—manual massage.

**Table 2 medicina-60-00125-t002:** The results of the mixed model and repeated measures ANOVA.

Interaction of Recovery Procedures	Wilks’ Lambda	F	Sig.	Eta Squared
**Passive rest**	0.432	4.831	0.022 *	0.568
**Percussive mechanical massage**	0.302	8.486	0.003 **	0.698
**Vibro-mechanical massage**	0.311	8.112	0.004 **	0.689
**Manual massage**	0.504	3.611	0.049 *	0.496
**Recovery procedure × Measurement time point**	0.873	0.778	0.637	0.044

* *p* < 0.05, ** *p* < 0.01.

**Table 3 medicina-60-00125-t003:** The results of the Bonferroni post hoc test—pairwise comparison.

		Passive Rest		Percussive Mechanical Massage		Vibro-Mechanical Massage		Manual Massage	
		Tc	Dm	Tc	Dm	Tc	Dm	Tc	Dm
**BL**	**PF**	0.049 *	0.002 **	0.049 *	0.004 **	0.014 *	0.022 *	0.042 *	0.047 *
	**PRC**	0.008 **	0.002 **	0.001 **	0.018 *	0.043 *	0.027 *	0.030 *	0.148
	**PRT**	0.028 *	0.005 **	0.001 **	0.009 **	0.399	0.105	0.287	0.205
**PF**	**PRC**	1	1	1	1	1	1	0.739	0.803
	**PRT**	1	0.719	1	1	0.028 *	0.246	0.45	1
**PRC**	**PRT**	0.335	1	0.174	1	0.121	0.199	1	1

Tc—muscle contraction time, Dm—maximal radial muscle displacement, BL—baseline measurement, PF—post-fatigue measurement, PRC—post-recovery measurement, PRT—5-min post-fatigue measurement; * *p* < 0.05, ** *p* < 0.01.

## Data Availability

The datasets obtained and analyzed for the current research can be provided by the authors of this study.

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
