# Peer review of "Acute Effects of Short-Term Massage Procedures on Neuromechanical Contractile Properties of Rectus Femoris Muscle"

_medicina, 2024, doi:10.3390/medicina60010125_

Round 1

Reviewer 1 Report

Comments and Suggestions for Authors

Abstract

In general, this summary responds very well to what has been done in this research. The different parts are distinguished, the objective is well described, how the research has been carried out, what the main results have been and finally what is the main conclusion reached by the authors.

In line 21 when you include the initials TMG please indicate what this acronym stands for, you do it very well for Dm and Tc in line 25 but you have not done it for the initials TMG. Please correct this.

The abstract is sufficiently attractive for the future reader to be able to continue reading the rest of the article and potentially be cited in future works.

1. Introduction

Lines 40-45. The different statements you include in this piece of text are the basis on which this research is based. They are varied and of different orientation: sport success, fatigue-stress, passive recovery, mucle function ..... But you only include a single quote (1) from 2005. Please reconsider this and enrich it with many more and much more current quotes.

Lines 45-55. You should update the citations... you even present some from the 90's !!!!

Lines 56-71. Ok, good literature review

Ok, good identification of objectives and hypothesis.

2. Materials and Methods 

2.1. Ethical approval

Ok, good work

2.2. Participants

Ok

2.3. Experimental protocol

Ok, good explanation of the protocol

2.4. Experimental procedures 128

2.4.1. TMG measurements

Ok, good explanation

2.4.2. MVIC, warm-up and fatigue procedures

Ok, good explanation

2.4.3. Recovery treatments

Ok, good explanation

2.5. Statistical analysis

Very well explained and very well selected statistical treatment.

3. Results

At the bottom of the page, on line 253, there are two unlisted graphs. Please correct this.

4. Discussion

This is undoubtedly the best section of the entire article in which the authors have made a real effort to compare the findings of their research with those of previous studies. Congratulations

5. Conclusions

Please include here some clear and straightforward recommendations for athletes and coaches to apply the new knowledge generated by this research.

Reviewer 2 Report

Comments and Suggestions for Authors

Dear Authors,

First of all, I would like to congratulate the authors for their efforts in developing this research. The research is interesting, although with limited clinical impact, but the manuscript presents formal limitations and small methodological errors.

ABSTRACT:
The abstract uses abbreviations that are not described (they should be eliminated since they are not recommended in this section).

INTRODUCTION:
Other research has evaluated the immediate effect of other Physical Therapy and Physical Medicine techniques and methods (e.g., doi: 10.3390/biology12030454). I believe the Authors should properly contextualize this.

METHODS:
Zeros as the last decimal place mean nothing, please remove them.

RESULTS:
Descriptive Results it is sufficient to convey them with only one decimal place.

DISCUSSION:
Correct. Congratulations.

Kind regards

The research addresses the Acute Effects of Short-Term Massage Procedures on Neurome- 2
chanical Contractile Properties of Rectus Femoris Muscle. This topic is interesting because in the available scientific literature there are hardly any bibliographic references that address the immediate, short-, medium-, long-term and very long-term effects. It is important that Physical Therapy and Physiotherapy professionals base their therapeutic actions on Evidence Based Medicine. And this work actively participates in the achievement of that objective.

It follows from the above that the research (and its object of study) are interesting for the community and novel.

Overall, the manuscript is well written. Although some formal errors have been identified and the Authors will be able to find them specifically within my comments.

The Conclusions are adequate with the stated objectives and the applied methodology.
